# Water Footprint of Wheat in Iraq

**Salam Hussein Ewaid** [1] , **Salwan Ali Abed** [2] **and Nadhir Al-Ansari** [3,*]

1    Technical Institute of Shatra, Southern Technical University, Al-Qadisiyah 58001, Iraq;
     salamalhelali@stu.edu.iq
2    Department of Environment, College of Science, University of Al-Qadisiyah, Al-Qadisiyah 58001, Iraq;
     Salwan.abed@qu.edu.iq
3    Luleå University of Technology, 97187 Luleå, Sweden
*    Correspondence: nadhir.alansari@ltu.se

**Abstract:** The water footprint (WF) is an indicator of indirect and direct fresh water use. In respect of facilitating decision-making processes, WF gives an excellent perspective on how and where fresh water is used in the supply chain. More than 39 million people live in Iraq and, with a growing population, there is a water shortage and a rising demand for food that cannot be met in the future. In this study, the WF of wheat production is estimated for the year 2016–2017 for 15 Iraqi provinces. The WF was calculated using the method of Mekonnen and Hoekstra (2011) and the CROPWAT and CLIMWAT softwares' crop water requirement option. It was found that the WF in $m^3$/ton was 1876 $m^3$/ton. The 15 provinces showed variations in WFs, which can be ascribed to the difference in climate and production values. The highest wheat WF was found in Nineveh province, followed by Muthanna, Anbar, and Basra. The last three provinces produce little and have a high WF so, in these provinces, wheat can be replaced with crops that need less water and provide more economic benefit. There is an opportunity to reduce the green WF by increasing production from the 4 rain-fed provinces, which will reduce the need for production from the irrigated provinces and, therefore, reduce the use of blue water.

**Keywords:** water footprint; wheat; CROPWAT; Iraq

## 1. Introduction

Agriculture is the principal user of fresh water around the world, accounting for almost 70% [1] of water supply.

With the effects of both socioeconomic development and climate change, the water crisis has turned into a problem throughout the world [2]. Even though water is a sustainable resource, access to it differs both spatially and temporally, and the gap between growing demand and restricted water resources is increasing [3]. The climate of Iraq has semi-dry features but, until a few decades ago, the country was rich in water [4]. Climate change has increased the need for water [3], and the construction of many dams on the Euphrates and Tigris Rivers in Turkey, Syria and Iran has caused water shortages downstream. Today, there is a need for a reduction in consumption, good planning of water resources, and a way to determine the water requirements of the main crops [5].

Land suitable for agriculture in Iraq accounts for less than 15% of the country's total area, with only 4 to 5 million hectares of the available 8 million being cultivated [6].

In the past, irrigation water resources in Iraq were not managed efficiently. However, this did not pose major problems because of the plentiful supply of water, albeit dependent on the Euphrates and Tigris Rivers and reasonable rainfall levels. Now, however, with the water scarcity crisis, increased salinity, low rainfall levels and a decrease in the discharge from rivers, there is a need to improve irrigation systems in order to resolve water problems [7–9]. These problems are only likely to become

more prominent in the future, with the supply 43 billion cubic meters (BCM) in 2015 and anticipated to be 17.6 BCM in 2025, while current demand is somewhere in the range of between 66.8 and 77 BCM [10].

The idea of the water footprint (WF) was presented by [11] and offers another way to evaluate the use of water resources in food production. WF accounting can also be a useful tool to measure and predict the consumptive water use in rain-fed and irrigated agriculture in order to meet demand through either rainwater or irrigation, or a combination of both [12].

The WF for a crop is described as the volume of fresh water that is utilized during all process of production [11]. WF has three forms: green, blue and gray. Blue WF refers to the consumption of groundwater and surface water during the supply chain of the product, green WF refers to the consumption of rainwater before it becomes run-off, and gray WF is the necessary volume of water required to convert polluted water into an acceptable standard of water quality [13]. WF accounting can be done at catchment, subnational, national, and international level, and it can be assessed from a consumer or producer perspective [12]. Likewise, it can be measured for a particular food crop, as has been done in this paper.

The WF of a crop product is generally estimated in two different ways: the total WF in a particular location (in m$^3$), or the WF of the product unit mass (in m$^3$/ton). The total WF is directly related to water resource accessibility; the blue and green WFs of unit production describe the local water productivity [14].

Bread wheat, *Triticum aestivum* L, is the most important crop in Iraq as bread is a major staple foodstuff. Despite this, annual bread wheat production in the country fluctuates and is constrained by factors such as low rainfall levels, salinity, and local conflicts. The cultivated area for wheat in Iraq is about 1,675,427 million ha and, in the 2016–2017 agricultural seasons, the yield was 2,195,574 million tons. However, production levels are insufficient to meet demand and, consequently, Iraq imports 3–4 million tons each year [15].

The majority of wheat production in Iraq relies on irrigation water, and only 39.7% of the wheat cultivation area is rain-fed. The areas for rain-fed wheat production are located in the north of Iraq, where the climatic conditions favor such cultivation requirements because Iraqi wheat is cultivated during November and harvested during May [16].

Iraq accounts for 0.4% of global wheat production and is ranked 31st globally, with a production of 5,055,111 tons in 2014, 2,645,061 tons in 2015, and 3,052,939 tons in 2016 [17]. In previous years, the bulk of wheat production was from the three northern Iraqi provinces (Nineveh, Kirkuk, and Saladin). The province of Nineveh, which was traditionally the main producer of wheat but where, during the years covered by this study, production had declined because of the conflict caused by the Islamic State in Iraq and Sham (ISIS). Before the conflict, Nineveh province was producing 21% of Iraq's total output [18,19]. It should be noted that wheat fields in the Iraqi Kurdistan Region occupy about 567,625 ha and produce an estimated 500,000 tons each year [20]. However, this region's figures were not included in this study due to insufficient data.

For 2018, the wheat production estimate shows a 14% decrease on the previous year's level and an approximate 20% fall when compared to the previous five-year average [15].

In the year covered by this study, the southern province of Wasit, which has irrigated land, ranked first in wheat production by 537,982 tons out of a 4,120,160 tons total production [15].

In the past two decades, many studies have been published on the WF of wheat production. Refs. [21,22] assessed the WF of wheat by analyzing blue and green water for the main producers around the world. Refs. [21,22] also evaluated the blue, green and gray WF of wheat in Italy. Ref. [23], in their overview, noted the blue, green and gray WF of some worldwide crops and their derived products, including wheat. Ref. [24] also studied the green and blue WF of irrigated and rain-fed wheat in the Sudan. Ref. [25] evaluated the WF of wheat and its derived products in Sweden. Ref. [26] calculated the WF of grain crops in characteristic irrigation regions of China using the CROPWAT model to calculate the evapotranspiration. Ref. [27] estimated the WF of wheat produced in Iran. Ref. [28] estimated the WF of major crops and vegetables in South Africa.

This study focuses on the WF of wheat by estimating the green and blue WF from a production perspective for both irrigated and rain-fed lands in Iraq in view of the actual water use by wheat production at the provincial scale. Total water use in each province of Iraq was also calculated. However, gray WF is excluded from this study due to lack of data.

The wheat crop was chosen due to its importance in Iraq and to the availability of data over the 2016−2017 period. The WF estimates in this research can provide benchmarks for the country and can conceivably prompt policymakers towards maintainable and economically efficient water administration strategies.

## 2. Method and Data

### 2.1. The Study Area

Iraq is located between latitudes (37.381096° and 29.057532°) and longitudes (38.793229° and 48.660797°). It is part of West Asia, has an area of 438,317 sq km and consists of 18 provinces (Figure 1). The geography of Iraq is diverse, and it has four main regions: 1. The northern highlands of Iraq; 2. Upper Mesopotamia; 3. Lower Mesopotamia, the alluvial plain spreading from Tikrit to the Arabian Gulf; and 4. the desert (west of the Euphrates) [29].

Wheat is cultivated throughout the country, but the better arable lands are mostly found in the north and northeast, where winter crops (barley and wheat) are grown along the river valleys. Cereal production accounts for 70% to 85% of the cultivated area [16].

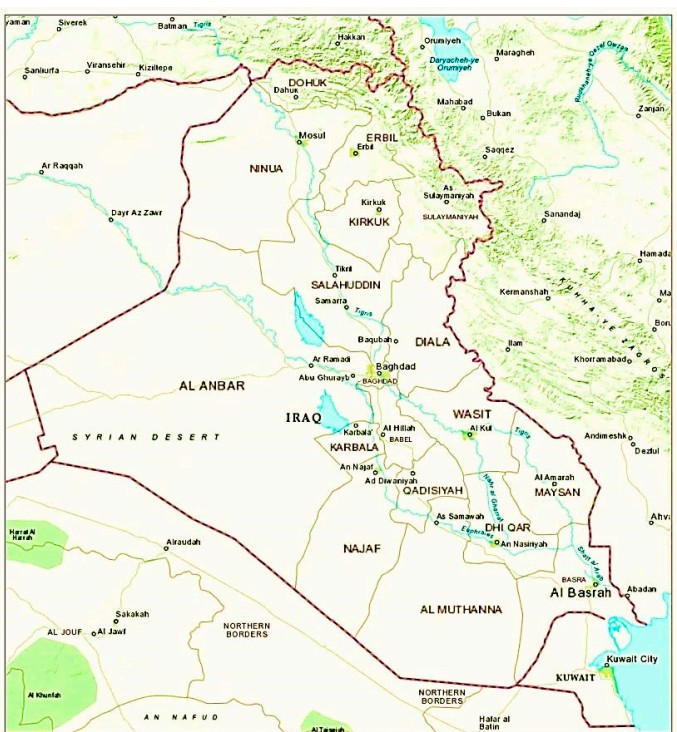

**Figure 1.** Map of Iraq and its provinces (the study area).

### 2.2. CROPWAT Model

The technique of [21,22] was followed for WF calculation. The blue and green WF calculations for wheat are described based upon the CROPWAT model method in view of [30].

The CROPWAT model is a decision support program advanced by the Food and Agriculture Organization of the United Nations (FAO). It is used for the calculation of irrigation water requirements (IWR) and crop water requirements (CWR) based on climate, soil, and crop data in view of the FAO publication [30].

In this study, the CWR method is used to calculate crop evapotranspiration (ET) in the CROPWAT model under ideal growth situations, which means that suitable soil water is preserved by rainfall and/or irrigation.

Four types of data are essential for using the CWR method with the CROPWAT model: climate, rainfall, crop, and soil data. Climate data for the period (1971–2000) was received from 12 Iraqi meteorological stations (listed in Table 1), obtained from CLIMWAT 2.0, which is a climatic database that can be used in relation with the CROPWAT model [31]. The CROPWAT climate and reference evapotranspiration ($ET_0$) data were used to calculate the average of ET0 per month based upon the FAO method from Penman-Monteith [30]. The rain data in CLIMWAT was used to calculate the effective rainfall (Eff.rain) using the United States Department of Agriculture (USDA) S.C. method [30]. The input crop data located in CROPWAT includes a crop coefficient (Kc), stages, rooting and depth. The status of soil in Iraq was considered as medium [31], and the planting and harvesting dates were taken as November and May, respectively [32].

*2.3. Crop Water Requirement (CWR)*

The volume of water needed by a crop to grow is called the crop water requirement (CWR, mm). The CWR level varies spatially and temporally, and also according to the type of crop. It is influenced by two factors: the crop coefficient (Kc) and the reference crop evapotranspiration ($ET_0$, mm). These two factors are affected by climate variants, such as wind speed, sunshine, temperature, and humidity [21,22].

The CWR is calculated by multiplying $ET_0$ by the Kc and is equal to the actual crop evapotranspiration (ETc, mm) under ideal conditions with no water limitations to crop growth under ideal conditions with no water limitations to crop growth.

$$CWR = Kc \times ET_0 \tag{1}$$

$$CWR = ETc \tag{2}$$

$ET_0$ is the evapotranspiration rate from a reference surface, without any scarcity of water.

The reference crop is a theoretical grass reference crop with specific features, and only the climatic parameters will influence it [30].

Kc is a value that differentiates field crops from the reference grass crop used for $ET_0$.

Variations in the Kc occur because of the change in crop features during crop growing and are influenced by crop variety, climate, and growth stages. The crop-growing period is divided into four stages: the initial, the development, the mid-season and the late season stage [21,22].

*2.4. Crop Water Use (CWU)*

Estimating the crop water use (CWU) requires estimations of ET rates for the studied crop in the climate of the region. This site-specific data was obtained from the CROPWAT 8.0 and CLIMWAT 2.0 modelling software. CROPWAT uses CLIMWAT rainfall data, soil data, and crop growth inputs to calculate the CWR under ideal conditions during the entire growth period [31].

2.4.1. Green CWU

The green quantity of crop water use (CWUgreen, $m^3$/ha) is the amount of rainwater that the wheat crop uses during evapotranspiration. It is calculated by gathering the daily green evapotranspiration (ETgreen, mm/day) during the entire growth period. A factor of 10 is used to change the water depths in mm to water volumes in $m^3$/ha. The summation is done using time advances of 10 days over the length of the entire development time of a crop [30].

$$CWUgreen = 10 \times \sum_{d=1}^{Igp} ETgreen \tag{3}$$

The ETgreen is either the effective rain (Eff.rain) or the ETc. If the Eff.rain is more than the CWR, the ETgreen will be equal to the ETc value because a crop never utilizes more than is required for ideal growth. If the Eff.rain is smaller than the ETc, the ETgreen will be the total Eff.rain [21,22].

$$\text{ETgreen} = \min \left( \text{ETc, Eff.rain} \right) \tag{4}$$

### 2.4.2. Blue CWU

The blue constituent of CWU (CWUblue, $m^3$/ha) is the amount of crop irrigation water needed for growth, including ground and surface water.

$$\text{CWUblue} = 10 \times \sum_{d=1}^{\text{Igp}} \text{ETbule} \tag{5}$$

The ETblue, otherwise called the irrigation requirement (IR), is the difference between the ETc and the Eff.rain. If the Eff.rain is more than the ETc, the ET blue is zero, and thus no irrigation is needed. If the CWR is not completely met by Eff.rain at that time, the ET blue is the difference between them.

$$\text{ETblue} = \max \left( 0, \text{ETc, Eff.rain} \right) \tag{6}$$

### 2.5. Total Water Footprint (WF) of a Crop

The total WF of a crop is the summation of the green (WFgreen, $m^3$/ton) and blue (WFblue, $m^3$/ton). It provides the amount of water used in order to produce a certain crop yield in $m^3$/ton [21,22].

$$\text{WF} = \text{WFgreen} + \text{WFblue} \tag{7}$$

The WFgreen is computed by dividing the green CWU ($m^3$/ha) by the crop yield (Y, ton/ha).

$$\text{WFblue} = \frac{\text{CWUgreen}}{\text{Y}} \tag{8}$$

The WFblue is computed by dividing the blue CWU ($m^3$/ha) by the crop yield (Y, ton/ha).

$$\text{WFblue} = \frac{\text{CWUblue}}{\text{Y}} \tag{9}$$

The WF crop production refers to the volume of water used to develop the amount of crop for a specific period. The production total WF of a crop ($m^3$/year) is the summation of the blue, and green WF components by $m^3$/year.

Computations for the two WF crop production constituents are done by multiplying the annual production (ton/year) by the water footprint per unit mass of crop ($m^3$/ton), and the WF of each component will be shown in terms of the volume of cubic meters of water per time period ($m^3$/year).

$$\text{WFgreen} = \text{WFgreen} \times \text{production} \tag{10}$$

$$\text{WFblue} = \text{WFblue} \times \text{production} \tag{11}$$

### 2.6. Data Requirement

The climate data for the period (1971–2000) was gathered from 12 Iraqi climatological stations (Table 1) by the CLIMWAT 2.0 climatic database used in association with the CROPWAT thus allowing for calculation of the CWR for different crops near many of the meteorological stations around the world [31]. The 12 stations' data contains station name, country, longitude and altitude, each one represents the climate state for one or two provinces due to the lack of stations and the fact that climate variation between the provinces in Iraq is small.

CLIMWAT 2.0 contains long-term monthly averages for the seven climatic parameters, monthly maximum and minimum temperature (°C), wind speed (km/h), mean relative humidity (%), sunshine hours (h), rainfall data (mm), and effective rainfall (mm) [33].

The data for the wheat crop's Kc values per province were obtained from the CROPWAT crop database based on [30,31], including rooting depth, crop coefficient, critical depletion, yield response factor, and length of plant growth stages.

The local data on crop yield (ton/ha) and production (ton/year) were obtained from the Iraqi Ministry of Agriculture, Office of the Inspector-General [15]. Wheat planting dates were taken according to the Iraqi Department of Extension and Agricultural Training's guide to agricultural operations [32].

The soil parameters obtained from the FAO CROPWAT 8.0 model included detailed information on the soil near the climatic station, such as total available moisture content, initial moisture depletion, maximum rain infiltration rate and maximum rooting depth. The USDA soil conservation method was used. The status of soil in Iraq was considered as a medium [31].

## 3. Results and Discussion

The results of the current study are introduced in the following categories:

### 3.1. General Features of the Climate in Iraq

The climate of Iraq is of a continental, semi-arid subtropical type. Rainfall occurs from November to April, and the average annual rainfall is about 216 mm. Winter is cold, with the daytime temperature about 16 °C, falling to 2 °C at night. Summer is hot and dry, with a shade temperature of over 43 °C in July and August, decreasing at night to 26 °C [34].

Tables 1 and 2 show results from the use of CROPWAT and CLIMWAT software and describe the 12 weather stations and related 15 provinces included in this research together with the spatial-temporal variation of climatic data from the stations, their positions, and altitude. The distribution of rain was relatively low in the southern region and more in the north.

The average annual rain for the 12 stations ranged from 96 mm in Najaf, which received the lowest amount of rain, to 441 mm in Kirkuk, which received the highest rain amount. Across the whole country, January has the most rainfall, followed by December.

**Table 1.** Some characteristics of the 12 meteorological stations.

| Station | Province | Altitude (m) | Latitude North | Longitude East | ETblue (mm) | ETgreen (mm) | Annual Rain (mm) |
|---|---|---|---|---|---|---|---|
| Basra | Basra | 2 | 30.56 | 47.78 | 113.4 | 96.1 | 134 |
| Amara | Misan | 9 | 31.85 | 47.16 | 87.2 | 121.4 | 168 |
| Nasiriya | Dhi-Qar | 3 | 31.8 | 46.23 | 216.5 | 67.8 | 107 |
| Semawa | Muthanna | 6 | 31.3 | 45.26 | 168.7 | 77.7 | 109 |
| Kut | Wasit | 15 | 32.16 | 46.05 | 165.6 | 89.4 | 151 |
| Najaf | Najaf and Karbala | 32 | 31.98 | 44.31 | 218.3 | 56 | 96 |
| Diwaniya | Qadysia | 20 | 31.98 | 44.98 | 202.5 | 75.6 | 116 |
| Baghdad | Baghdad and Babylon | 34 | 33.23 | 44.23 | 138 | 98.7 | 149 |
| Ana | Anbar | 150 | 34.46 | 41.95 | 103.5 | 78.4 | 122 |
| Kanaqin | Diyala | 202 | 34.3 | 45.43 | 2.1 | 209.8 | 317 |
| Kirkuk | Kirkuk and Saladdin | 331 | 35.46 | 44.4 | 0 | 259.4 | 441 |
| Mosul | Nineveh | 223 | 36.31 | 43.15 | 0 | 241 | 421 |

The average values of the minimum and maximum air temperature were 15.35 °C and 30.26 °C. Average wind speed was 209 km/day. The daily average $ET_0$ was 5.38 mm/day. The total average effective rainfall was 126.7 mm/dec (Table 2). The $ET_0$ variable did have a small variation in the study region for the studied period, but there are some differences in ETc values among the northern stations (Ana, Kanaqin, Kirkuk, and Mosul) and the other 8 stations in the middle and south. In the four northern stations, the green evapotranspiration (ETgreen) is more than in the other eight stations. The ET blue in some southern stations like (Nasiriyah, Najaf, and Diwaniya) was higher compared to the other stations (Table 1). The average humidity was 45.25% for all the stations. The average irrigation requirement was 123.51 mm/dec (Table 2).

### 3.2. Crop Water Requirements

Data about variation in CWR is important for the accurate calculation of the crop WF. The CROPWAT FAO model [30,31] was used to evaluate the blue (irrigation) and green (effective rainfall) consumptive water use of the wheat crop.

CROPWAT estimated the Eff. rain as well as the ETc, considering climate, crop, soil parameters, and daily soil profile moisture status. Tables 1 and 2 show the 30-year (1971–2000) average climatic data obtained by CLIMWAT that was used for the CWR computation.

The average ETc was 227.1 mm, the IWR was 123.51 mm and the Eff.rain was 126.7 mm/dec. The Eff.rain values and ETc of wheat grown in the 15 Iraqi provinces showed important local variation (Table 2). This may be because of the spatial variations in rainfall and temperature patterns. The ETc values for wheat in the different provinces varied from 139.4 mm to 284.4 mm, and this regional difference should be measured in the calculation of wheat WF. The increase in CWR was due to the hot and sunny climate as the crop needs more water than in a cool and cloudy climate. Although crop yield is good in the irrigated regions, ETc is higher in these regions as well. In rain-fed regions, the ETc during the growing period is low while, in irrigated situations, there is more water accessible to meet CWR, leading to high ETc.

**Table 2.** Data for evapotranspiration (ETc), effective rainfall (Eff.rain), irrigation water requirements (IWR) and related variables in the study area for the period from (1971 to 2000) obtained by the CROPWAT and CLIMWAT software.

| Station | ET0 (mm/d) | ETC (mm/dec) | Eff.rain (mm/dec) | IWR (mm/dec) | Humidity (%) | Wind (km/day) | Sun (h) | Temp. °C Min | max |
|---|---|---|---|---|---|---|---|---|---|
| Basrah | 4.81 | 209.5 | 96.1 | 120.3 | 64 | 205 | 8.1 | 17.8 | 30.4 |
| Amarah | 5.57 | 208.5 | 121.4 | 97.4 | 46 | 197 | 8.2 | 16.4 | 31 |
| Nasiriya | 6.23 | 284.4 | 67.8 | 216.2 | 44 | 259 | 8.1 | 16.6 | 31.6 |
| Semawa | 5.73 | 246.4 | 77.7 | 171.6 | 38 | 204 | 8.1 | 16.4 | 31.3 |
| Kut | 5.62 | 255 | 89.4 | 165.8 | 46 | 215 | 8.3 | 16.2 | 31.5 |
| Najaf | 6.11 | 274.3 | 56 | 219 | 39 | 241 | 8.3 | 16.8 | 31.1 |
| Diwaniya | 5.68 | 278 | 75.6 | 202 | 44 | 241 | 8.5 | 14.7 | 31 |
| Baghdad | 5.59 | 226.7 | 98.7 | 128.4 | 45 | 220 | 8.2 | 14.7 | 30.4 |
| Ana | 4.92 | 182 | 118.4 | 108.5 | 44 | 179 | 7.9 | 13.6 | 28.3 |
| Kanaqin | 4.8 | 212 | 210 | 42.4 | 36 | 158 | 7.6 | 14.7 | 30 |
| Kirkuk | 4.86 | 190.3 | 259.4 | 9.2 | 42 | 184 | 8 | 15.3 | 28.4 |
| Mosul | 4.72 | 139.4 | 250 | 1.4 | 55 | 194 | 7.7 | 11.1 | 28.2 |
| Average | 5.38 | 227.1 | 126.7 | 123.51 | 45.25 | 208 | 8.08 | 15.35 | 30.26 |

### 3.3. Wheat Water Footprint (WF)

The total water used for wheat cultivation in the 15 Iraqi provinces during 2016–2017 was 4,163,347,053 m$^3$/year on average, the major share (60.66%) coming from green water and about 39.34% coming from blue water. About (63.56%) of the wheat produced in Iraq was irrigated, and 36.44% was rain-fed from about 1,675,427 ha of wheat-cultivated lands.

The spatial variation of water use for wheat production is shown in Table 3. Blue water use varied from zero in the rain-fed provinces to 2184 m$^3$/ha in Najaf. The green water parts are more

than half of the utilized water; this means that wheat production was mainly dependent on green water. Green water use is found in all the provinces, but the blue water use is zero for the four rain-fed provinces (Diyala, Kirkuk, Saladin, and Nineveh).

The green, blue and total WF of wheat over the agricultural season 2016–2017 in the 15 Iraqi provinces was calculated using the method of [11,30]. For each province, the green water footprint (m$^3$/ton) has been evaluated as the proportion of the green water utilized (m$^3$/ha) to the crop yield (ton/ha), where total green water utilization is figured by summing up green water evapotranspiration during the growing period. The blue WF has been taken as being equivalent to the proportion of the consumed irrigation water by the wheat yield. The total WF is the summation of green WF and blue WF.

The parts of wheat WF were evaluated for the 15 provinces (Table 3, Figure 2). For irrigated regions, the green WFs fluctuated from 238 in Najaf to 2068 m$^3$/ton in Anbar, and the blue WFs from 614 in Misan to 2960 m$^3$/ton in Muthanna.

For rain-fed regions, the green WFs fluctuated from 1076 in Kirkuk (the lowest in the country) to 5477 m$^3$/ton in Nineveh province, which has the highest overall WF in the country. Babylon province has the smallest WF among the irrigated area, as shown in Table 3 and Figure 2.

To calculate the national footprint of the country, each footprint (m$^3$/ton) of a province was multiplied by the quantity of wheat produced there. The output of the provinces was collected and divided by the total crop produced in the country (2016–2017), giving a figure of (1876 m$^3$/ton), which represents the water footprint of the wheat crop in Iraq.

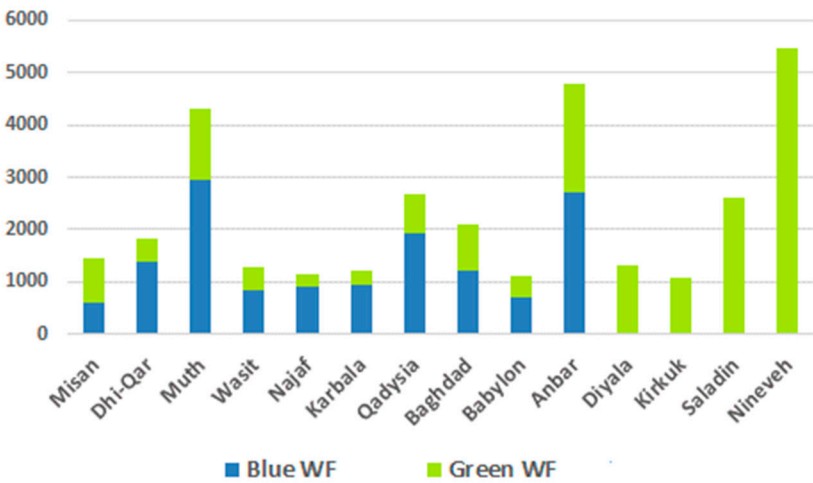

**Figure 2.** Components of wheat water footprint (WF) for the 15 Iraqi provinces (m$^3$/ton).

Given the above results, it might be expected that Iraqi provinces are a good production zone for wheat since they have lower WFs. The green WF is high since the greater part of the zone is under rain-fed conditions to varying degrees. The management of green water utilization has a key role, therefore, in economical water management and worldwide food security. Green water use in agriculture is vital since it makes rainfall more beneficial by increasing agricultural production. It is also important to note that, in rain-fed systems, there is no correlative blue water utilization decrease to accompany an expansion in green water flow. An expansion in green water utilization (ET green) may result in a reduced green WF if there are adequate yield increments alongside the expanded ET. Consequently, with the end goal being to lessen WF, a no-tillage approach can be embraced as there might be an expansion in the infiltration of precipitation and a decrease in non-useful soil evaporation. Adding a high residue cover crop that gradually expands soil organic matter may expand ET while really decreasing the WF due to yield increase.

The Iraqi WF per ton of wheat obtained in this study is higher than the world average (1622 m$^3$/ton), but less than in neighboring countries (Iran, Turkey, Syria), and less than that of international producers (Russia, and the United States) but more than in China, Egypt, and India (see Table 4) [35–37].

**Table 3.** Calculated wheat WF in Iraqi provinces during (2016–2017).

| Province | Crop Water Use (m³/ha) | | | Total WF of Production (Mm³/year) | | | Planted Area | Production | | Water WF per ton of Wheat (m³/ton) | | | |
|---|---|---|---|---|---|---|---|---|---|---|---|---|---|
| | Blue | Green | Total | Blue | Green | Total | ha | ton/yr | ton/ha | Blue | Green | Total | Position |
| Basra | 1134 | 861 | 1995 | 20,793,024 | 15,787,296 | 36,580,320 | 18,336 | 9956 | 0.54 | 2100 | 1780 | 3880 | 4 |
| Misan | 872 | 1214 | 2086 | 81,163,144 | 112,995,478 | 194,158,622 | 93,077 | 131,973 | 1.42 | 614 | 855 | 1469 | 9 |
| Dhi-Qar | 2165 | 678 | 2843 | 152,500,435 | 47,757,642 | 200,258,077 | 70,439 | 111,347 | 1.58 | 1390 | 429 | 1799 | 8 |
| Muthanna | 1687 | 777 | 2464 | 65,052,407 | 29,961,897 | 95,014,304 | 38,561 | 22,201 | 0.57 | 2960 | 1363 | 4323 | 3 |
| Wasit | 1656 | 894 | 2550 | 451,723,680 | 243,865,320 | 695,589,000 | 272,780 | 537,982 | 1,97 | 835.5 | 454 | 1289 | 11 |
| Najaf | 2184 | 560 | 2744 | 137,019,792 | 35,133,280 | 172,153,072 | 62,738 | 147,409 | 2.35 | 929 | 238 | 1167 | 13 |
| Karbala | 2183 | 560 | 2743 | 27,189,265 | 6,974,800 | 34,164,065 | 12,455 | 28,297 | 2.27 | 962 | 247 | 1209 | 12 |
| Qadysia | 2025 | 756 | 2781 | 244,543,050 | 91,296,072 | 335,839,122 | 120,762 | 126,469 | 1.04 | 1947 | 727 | 2674 | 5 |
| Baghdad | 1380 | 987 | 2367 | 82,700,640 | 59,148,936 | 141,849,576 | 59,928 | 68,234 | 1.13 | 1221 | 873.5 | 2094 | 7 |
| Babylon | 1380 | 987 | 2367 | 103,500,000 | 74,025,000 | 177,525,000 | 75,000 | 189,985 | 2.53 | 723 | 390 | 1113 | 14 |
| Anbar | 1035 | 784 | 1819 | 57,924,810 | 43,877,344 | 101,802,154 | 55,966 | 21,647 | 0.38 | 2724 | 2068 | 4787 | 2 |
| Diyala | 21 | 2098 | 2119 | 2,732,940 | 273,033,720 | 275,766,660 | 130,140 | 208,874 | 1.60 | 13 | 1311 | 1324 | 10 |
| Kirkuk | 0 | 2594 | 2694 | 0 | 235,906,142 | 235,906,142 | 90,943 | 219,131 | 2.41 | 0 | 1076 | 1076 | 15 |
| Saladin | 0 | 2594 | 2594 | 0 | 55,6672,400 | 556,672,400 | 214,600 | 214,154 | 0.99 | 0 | 2620 | 2620 | 6 |
| Nineveh | 0 | 2410 | 2410 | 0 | 866,881,820 | 866,881,820 | 359,702 | 157,915 | 0.44 | 0 | 5477 | 5477 | 1 |
| Total | | 1,426,843,187 | 2,693,317,147 | 4,120,160,334 | 1,675,427 | 2,195,574 | | 34.65% | 65.35% | Total | | | |
| Average | 851.6 | 1607 | 2459 | 34.63% | | | 65.37% | | 1.375 | 650 | 1226 | | 1876 |

It was clear that the green WF of the national wheat production of rain-fed provinces in Iraq is 60.66% during 2016–2017 and the green WF is more than the blue WF, confirming the significance of green water in wheat production. Green water, by and large, has a lower opportunity cost compared to blue water [37,38]. Since wheat has generally low financial water profitability when contrasted with numerous different yields [39], in provinces with a large blue water footprint and relatively little production (Basra, Muthanna, and Anbar), other crops requiring less water should be replaced with wheat. There are still opportunities to lower the green WF by increasing production from the four rain-fed provinces, which will reduce the need for production from the irrigated provinces, and thus reduce blue water use.

**Table 4.** Comparison of the results of this study with the WF values of some countries in the world [37].

| Country | WF per ton of Wheat ($m^3$/ton) | | |
|---|---|---|---|
| | **Green** | **Blue** | **Total** |
| Iran | 2412 | 988 | 3400 |
| Russia | 2359 | 31 | 2390 |
| Turkey | 2081 | 131 | 2212 |
| USA | 1879 | 92 | 1971 |
| Syria | 1511 | 457 | 1968 |
| **Iraq (This study)** | **1226** | **650** | **1876** |
| India | 635 | 1173 | 1808 |
| World | 1279 | 343 | 1622 |
| China | 820 | 466 | 1286 |
| Egypt | 216 | 907 | 1123 |

## 4. Conclusions

The results from this study reveal that the green WF related to Iraqi wheat production is about double the blue WF, confirming the significance of green water in the production process. Green water, for the most part, has a lower opportunity cost when compared to blue water. Since wheat generally has a low monetary water efficiency when contrasted with different other crops, there is no need to waste water in the water-scarce provinces to produce an uneconomic crop. The yield in the rain-fed provinces shows that there is an opportunity to lower the green WF. Expanding production from the rain-fed provinces will decrease the requirement for production from the irrigated provinces in the water-scarce areas of the center and south of the country, and thus reduce blue water use.

**Author Contributions:** S.H.E. and S.A.A. collected the data, analysis, validation investigation, data curation, and writing—original draft preparation. N.-A.A. did the visualization, project administration, and writing the final draft.

**Funding:** This research received no external funding.

**Conflicts of Interest:** The authors declare no conflict of interest.

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
