# Peer review of "Water Footprint of Wheat in Iraq"

_water, doi:10.3390/w11030535_

Round 1

Reviewer 1 Report

 Q.1: In Abstract section, Line No.9. The author (s) mentioned that the 15 provinces showed a variety concerning the WF while in Results and Discussion, Figure No.2 shows only 14 provinces? 

Q.2: In Study area, line No.1. To be more clear, the GPS coordinates should be given in a formula of Degree Minutes Second DMS and also the map is unclear and need Sat-image to interpret the sampling site. Please clarify that. 
Q.3: What is the efficiency of the wheat production process in Iraq, and what recommendations can be made following this conducted research on the topic?
Q.4: In line No.4 under table.4. Since wheat is a low-value crop, why water is allocated to wheat production in Iraq?
Q.5: In conclusion section, What is the practical application of WF concept at the field scale ?

Author Response

Response to Reviewer 1

Thank you very much for your comments which improved our paper.

Please note the following:

1-      Figure 2 corrected

2-      The GPS coordinates and the map, corrected

3-      The efficiency of wheat production process in Iraq.

Agriculture and wheat production in Iraq are not efficient. There is a waste of water and not exploiting of large areas. Water resources management methods should be improved, and suitable crops should be grown in the appropriate province based on the availability of water.

4-      Since wheat is low value crop, why water is allocated to wheat production in Iraq?

Wheat is a daily staple food for all Iraqis and is cultivated by the ancient heritage of farmers in Mesopotamia. Plans should be made to educate farmers to grow crops economically appropriate according to the availability of water and the quality of soil.

5-      In conclusion section, what is the practical application of WF concept at the field scale?

In provinces that produce little wheat and consume much water, wheat must be replaced with a crop that consumes less water and has more economic benefit.

All land in the north of the country, which produces wheat depending on rain, must be exploited.

Reviewer 2 Report

An interesting paper.  A couple of corrections and clarrifications are needed in the Abstract.  The 3 in m3 needs to be superscripted throughout the abstract and the rest of the paper.

Line 14 - reference to Hoekstra but there is no corresponding reference in the reference list. 

Line 18 - reference is made to wheat production practices - but little is actually said about this in the paper.  Line 19 the world average water footprint is mentioned however this needs to be double checked against the information produced for wheat water footprints as it does not seem to match.  Lines 19 and 20 - a variety of countries are mentioned - but it is unclear how they fit - are these at the top and bottom end of the wheat waterfootprint or have they been selected at random?  Not surev why numbers are being put in brackets within the abstract.  A global overview of the wheat waterfootprint needs to be provided.

Lines 30-31 - does this really need to be said?

31-32:  Agriculture is a significant user of water - but needs to diferentiate between rain-fed and irrigated.

34-35 -  not sure what is meant by growing requests and restricted water assets.

35-36:  Water is a sustainable resource - not always - what about fossil groundwater?

37:  Climatologically is Iraq classified as sub-tropical?

38:  Is it climate change that is increasing the need for water or climate variability - specific evidence would be needed.

39:  whilst dams have been constructed in Turkey - what about Syria and Iran?

44-47 - somewhat unclear what is meant - English does need improving.  Mention of low rainfall - is this aridity or a product of drought?

48-50 - he figures do need to be explained more clearly.

68 Latin names should be in Italics.

71-72 - wheat production needs to be put into the wider perspective - what proportion is produced within country and what is imported?

77-78 - what is grown in the Iraqui Kurdistan Region has not been previously mentioned.

85:  unsure what is meant by 20% decay.

The discussion of the different provinces - 79-89 is somewhat confused - needs to make the situation much clearer in terms of differences in wheat production and farming practices.

102-103:  why is grey WF not common practice among the farmers - this needs to be clearly explained.

105:  mention of the period 2016-2017 yet 128 the climate data used is for 1971-2000 - am unsure how the water footprint for wheat grown in 2016-2017 can be calculated when the climate data go up to 2000.

110:  Latitude and longitude for Iraq - however these are specific points within the country rather than a latitudinal and longitudinal range for the country.

Quality of Figure 1 does need to be improved.

163:  why is a time advance of 10 days chosen?

174-187:  somewhat repetitive - the discussion of what is being done needs to be revisited to streamline the text and reduce repetition.

200-202 - repetition of what has been written earlier.

214-217 - some indication of the bseaonlatily of the climate is required especially considering that earlier it was stated that wheat was sown in November and harvesterd in May - no detailed discussion of this period is to be found, or indeed the variability in the climate parameters.

227-235:  Again repeating what has been said earlier.  Average windspeen in km/day is an unusual value to select.

Crop water requirements:  it might be better to consider presenting some of this information either in a table or as a graph.

256:  Not sure how the water footprint for wheat can be calculated for 2016-2017 when the meteorological data end in 2000.

It needs to be established early in the paper which provinces are relianton irrigation and which can sustain rainfed agriculture.

283-295:  Whilst some provinces might have high green WF, there area number with high blue WF as well.

289-290 - not sure what is meant here - let down by phrasing. 

298-299:  Not sure why the countries are listed - are these particularly high water footprints?

The issue of reliance on greenwater is ok - but the key issue is the variability of the rainfall - if this is becoming more variable then it brings into question the reliance on rain fed agrriculture in certain provinces.

A clear discussion of the climatic characteristics for the country needs to be made - specifically on the variability of rainfall in these provinces and especially during the growing season.

Author Response

Response to Reviewer 2

Thank you for your comments which improved our paper.

Please note the following:

The 3 in m is corrected in all the article.

Line 14: corrected, it is the same reference [23]

Line 18: This is included in the study, especially in the results tables

Line 19: We compare the Iraqi wheat WF obtained in this study (1876 m3/ton) with the wheat WF in the world and some other countries which are mentioned in the reference (37).

Line 19-20: Some countries neighboring Iraq with some important countries were selected, some of them more than the global average and some are less. The global overview of the wheat water footprint is summarized in Table 4.

Line 30-31: deleted

Line31-32: For both the rain-fed and irrigated.

Line 34-35: The word (assets) is replaced by (resources) to mean that: demand is increasing and resources are steady.   

Line 35-36: Groundwater needs a long time to be sustainable.

Line 37: corrected, the climate of Iraq is not (subtropical), its only semi dry.

Line 38: Drought and lack of rain is the most important manifestation of climate change in the area where Iraq is located, causing a shortage of water for agriculture   

Line 39: corrected, Turkey, Syria and Iran.

Line 44-47: It Is a comparison between the past and present. In the past, the poor management of the Tigris and Euphrates rivers was not clear because water was abundant. But now that quantity and quality have decreased, there is a need to improve water management.

Line 48-50: corrected

Line 68: corrected

Line 71-72: This sentence is added (The production is not enough and Iraq imports 3-4 tons each year)

Line 77-78: (as mentioned in) is deleted to be clearer

Line 85: Production fell 20 percent compared to the previous five years

Line 79-89: The discussion of the different provinces is corrected as possible.  

Line 102-103: It is corrected to mean (The use of fertilizers in the cultivation of wheat in Iraq is not common either for lack of availability or lack of experience).

Line 105: the climate data for 1971-2000 is used because this data is available free of charge within the CLIMWAT program for a relatively recent period (average 30 years) and has not undergone significant changes. The collection of accurate climatic information from 15 provinces is difficult.

Line 110: corrected.

Line 163: The CROPWAT is designed this way.

Line 174-187: Repetition has been reduced as much as possible.

Line 200-202: The earlier repetition has been reduced.

Line 214-217: modified.

Line 227-235: The (1971-2000) average wind speed was 208 km/day as calculated by CROPWAT (Table 2). The crop water requirements are shown in. Tables 1 and 2

Line 256: This is the only data that can be obtained as an average for a long period and is as close as possible to the present time.  We mentioned in the introduction which provinces are reliant on irrigation and which can sustain rainfed agriculture.

Line 283-285: It was mentioned

Line 289-290: Corrected

Line 298-299: Some countries neighboring Iraq and some important producing countries were selected from the references.

That's true and the farmer sometimes gambles because of climate variability.

The climate of northern Iraq differs from the climate of the center and south of the country, in the north the land is higher with less temperature and more rain.

Round 2

Reviewer 2 Report

First, I'd like to thank the authors for their detailed responses to my suggestions - these are very useful. 

Line 20-21 - the water footprint for wheat - useful to include, but it would help readers is the information is provided in two batches - the first showing the highest and lowest values from around the world - this provides a clear framework against which the Iraqi values can be compared.  The second batch could then be the footprint from neighboring countries - the detailed discussion of this should take place in the main body of the paper.  This strengthens the argument  for calculating the water footprint of wheat in Iraq and applies to lines 303-306.

Line 38:  It is important to provide a reference for evidence of climate change leading to increasing drought frequency/severity and a decline in rainfall - or is this climate variability causing the current changes, but climate change leading to more significant changes over the next 100 years.

Line 42 English needs improving eg Land suitable for agriculture in Iraq accounts for less than 15% of the country's total area, with only 4 to 5 million hectares of the 8 million available being cultivated.

Line 47:  I think it is important that the contents included in the comment is stated in the paper - it is a very important point.

Line 72:  should this be 3-4 million tons?

Figure 1:  I'm not sure if the clarity of the text will improve in the final version, but at the moment the text is slightly pixelated, and the map/text too small - with the reliance on the map to show the location of the provinces that many readers may be unfamiliar with this needs to be resolved.

Line 233:  Whilst CROPWAT produces wind speeds in km/day this is not a standard climatological/meteorological unit - it would be better to convert the value to either km/hour or metres per second - these are units more easily understood and allow easier comparison

Lines 303-306 - The comment response needs to be incorporated into the main text (see my first comments above)  An important issue to make clear is that the water footprints are for the entire countries and mask regional variations that would be found by differences in climates within large countries eg Russia, USA etc. 

Author Response

Dear Reviewer

Thank you very much for your comments.

Please note that we did the required changes as follow:

Line 20-21: These lines must be deleted from the abstract:

(Compared to other global studies the WF per ton of wheat was higher than the world average (1622 m3/ton) and more than in China, Egypt, and India but less than in Iran, Turkey, Syria, Russia, and the USA.).

These lines must be added instead of the 303-306 lines:

The Iraqi WF per ton of wheat obtained in this study is higher than the world average (1622 m3/ton), less than in neighboring countries (Iran, Turkey, Syria), and less than in the international producers (Russia, and the United States) but more than in China, Egypt, and India, Table 4 [35, 36, 37].

Line 38: the reference [3] added after:

 The climate change increased the need for water [3]

Line 42: Land suitable for agriculture in Iraq accounts for less than 15% of the country's total area, with only 4 to 5 million hectares of the 8 million available being cultivated [6].

Line 47: the lines 44-47 becume:

Before now, the irrigation water resources in Iraq were not managed efficiently, this issue was not demonstrating a really difficult because of the richness of water, depending on the Euphrates and Tigris Rivers with moderate rainfall levels, but now with water scarcity crisis, increase salinity, low rains levels and rivers discharge decrease, there is a need to improve the irrigation systems [7, 8, 9].

Line 72:  corrected (3-4 million tons).

Figure 1: corrected.

Line 233:

They use (km/dy) for wind speed in the FAO CROPWAT software method and its standard in the program.

Lines 303-306

These lines added instead of the 303-306 lines:

The Iraqi WF per ton of wheat obtained in this study is higher than the world average (1622 m3/ton), less than in neighboring countries (Iran, Turkey, Syria), and less than in the international producers (Russia, and the United States) but more than in China, Egypt, and India, Table 4 [35, 36, 37].

Thank you again.

Best regards.

Nadhir Al-Ansari
